# Pre-Transplant Total Lymphocyte Count Determines Anti-Thymocyte Globulin Exposure, Modifying Graft-versus-Host Disease Incidence and Post-Transplant Thymic Restoration: A Single-Center Retrospective Study

**DOI:** 10.3390/jcm12020730

**Published:** 2023-01-16

**Authors:** Antonio Giacomo Grasso, Roberto Simeone, Alessandra Maestro, Davide Zanon, Natalia Maximova

**Affiliations:** 1Institute for Maternal and Child Health—IRCCS Burlo Garofolo, Via dell’Istria 65/1, 34137 Trieste, Italy; 2Department of Transfusion Medicine, ASUGI, Piazza dell’Ospitale 1, 34125 Trieste, Italy

**Keywords:** hematopoietic stem cell transplantation, pediatric, anti-thymocyte globulin, graft-versus-host disease, thymus-dependent T-cell reconstitution, thymus size

## Abstract

The use of anti-thymocyte globulin (ATG) as part of conditioning to prevent graft-versus-host disease (GVHD) may severely impair immune reconstitution (IR). We analyzed relationships between ATG exposure, the recipient lymphocyte count, IR, and transplant outcome. We retrospectively reviewed patients aged ≤ 18 years who underwent allogeneic HSCT between April 2005 and April 2020. The outcomes of interest included the incidence of GVHD, overall survival (OS), and IR. IR was analyzed through thymic magnetic resonance imaging (MRI) and by quantifying T CD4+ and recent thymic emigrants (RTEs). The ATG-exposed group was split into a low ATG/lymphocyte ratio subgroup (ratio < 0.01) and a high ATG/lymphocyte ratio subgroup (ratio > 0.01). The low ratio subgroup had a higher incidence of GVHD (29 [59%] vs. 7 [16.6%]) but a better IR in both laboratory and MRI imaging assessments (*p* < 0.0001). The median thymic volume in the low ratio subgroup was significantly higher (14.7 cm^3^ vs. 4.5 cm^3^, *p* < 0.001). This was associated with a better OS and lower transplant-related mortality (TRM) (80.4% vs. 58.0%, *p* = 0.031) and (13.1% vs. 33.0%, *p* = 0.035). An individualized approach to ATG dosing allows for the obtainment of rapid thymic reconstitution and the best transplant-related outcomes.

## 1. Introduction

Despite the increasing success rate in hematopoietic stem cell transplantation (HSCT), graft-versus-host disease (GVHD) remains a significant burden in terms of mortality and morbidity [1,2]. Among the many strategies that have proved helpful in lowering the incidence and severity of GVHD, the primary one is prophylactic treatment with immunosuppressive therapy [3,4]. While the current literature documents the effectiveness of immunosuppressive therapy in reducing GVHD mortality, it also reveals the persistently high rates of relapse and infection-related mortality that influence the overall survival of HSCT recipients [5,6].

The addition of anti-thymocyte globulin (ATG) to conditioning therapy as a GVHD prophylaxis has become the standard of care for unrelated and mismatched donors. But it is still unclear how to optimize the dose and timing of ATG to minimize the delay in all classes of T-cell reconstitution and maximize the protective effect regarding GVHD and graft failure [7,8]. The data are mostly from studies that engaged adult patients using high ATG doses due to the greater burden of GVHD in the adult population [9,10,11]. ATG dosage is generally calculated per kilogram of body weight, though some studies have shown that the same dosage causes a nonidentical effect in different patients [12,13,14,15]. Moreover, it is still unclear what the anti-leukemic effect of ATG is and how a greater dosage could lead to a lower incidence of relapse [16].

Based on the known biological effect of ATG on T-cell chelation, certain studies have speculated that the use of the drug must be modulated according to the absolute lymphocyte count of the recipient. A high pre-transplant lymphocyte count would lead to the loss of a part of the ATG immunomodulation on the recipient’s lymphocytes due to impaired immunosuppression efficacy on donor lymphocytes and, consequently, lead to GVHD [17,18].

Studies investigating the relationship between ATG exposure and T-cell immune reconstitution or transplant outcomes are limited, especially in the pediatric cohort, where it is most relevant [15,17,19,20].

In this study, we retrospectively analyzed the outcomes for pediatric patients who underwent allogeneic HSCT and received prophylactic ATG, sorting patients into two groups based on the ratio of the total ATG dose and total lymphocyte count on the day of ATG administration.

The optimal reconstruction of the adaptive immune system after allogeneic HSCT is conditioned by thymic activity, as only active thymopoiesis leads to the reconstitution of a functionally competent, fully mature, naïve T-cell compartment with broad antigenic specificity [21,22].

We investigated thymus-dependent T-cell reconstitution by assessing the thymus size through magnetic resonance imaging (MRI) and quantifying recent thymic emigrants (RTEs) by immunophenotyping. Finally, the possibility of a relationship between thymic function and transplant-related outcomes, including the cumulative incidence of GVHD, overall survival (OS), event-free survival (EFS), and the recurrence rate, was analyzed.

## 2. Materials and Methods

A retrospective single-center study was carried out at the Pediatric Transplant Center of the Institute for Maternal and Child Health–IRCCS “Burlo Garofolo,” Trieste, Italy. The Institutional Review Board of the IRCCS Burlo Garofolo (reference no. RC 08/19) and the Unique Regional Ethics Committee (reference no. 2869) approved the study protocol. The study was retrospectively registered on ClinicalTrials.gov, number NCT04869254. All parents of the patients gave written consent for collecting and using personal data for research purposes.

We retrospectively reviewed clinical records of all patients aged ≤ 18 years who underwent allogeneic HSCT between April 2005 and April 2020 with ATG (Thymoglobulin, Genzyme Europe BV, Amsterdam, The Netherlands) as part of the myeloablative conditioning regimen (ATG-exposed group). Only patients undergoing their first HSCT were included. The inclusion criteria had no restriction on the indication, cell source used, and the dose of ATG infused. We excluded patients who received ATG the previous three months before HSCT or had an allergic reaction to ATG infusion. All patients received ATG in total dose ranging from 2.5 mg/kg to 12 mg/kg. These differences in dosage reflect systematic changes in institutional practice over the fifteen years covered by the study. All patients were treated in accordance with the standard of care protocols. The infusion of ATG was typically started four days before HSCT and given over three days. For transplant outcomes analysis, the patients who received ATG in conditioning were divided into two groups according to the ratio between the total ATG dose received (numerator), expressed in milligrams, and the lymphocyte count before the first ATG dose (denominator), expressed in cells × 10^2^/μL (ATG/lymphocyte ratio). To obtain two numerically homogeneous groups, we set 0.01 as the cut-off of the ATG/lymphocyte ratio.

All patients were treated according to standard myeloablative protocols, as previously described [23]. Additional GVHD prophylaxis included a calcineurin inhibitor and mycophenolate mofetil for the matched unrelated donor (MUD), with the addition of post-transplant cyclophosphamide from 2013 in the case of a haploidentical donor. The prevention and treatment of infection and other elements of transplant-specific supportive care were carried out according to institutional standard practices. The duration of follow-up was defined as the time from HSCT to last contact or death. The minimum follow-up for surviving patients was 12 months. Given the heterogeneity of patients, we also assessed the disease risk index by refined criteria, which consider disease status, stage, and cytogenetics [24]. The thalassemia disease risk index for allogeneic transplants was defined according to previously published criteria [25]. Ewing sarcoma with allogeneic transplant indication was considered to be very high risk.

The primary outcome assessed was the incidence of GVHD, defined as any GVHD requiring systemic immune suppressive therapy. Acute and chronic GVHD was defined by the date of first immune suppressive therapy occurring < 100 or ≥100 days after HSCT, respectively. Acute and chronic GVHD was diagnosed based on standard criteria [26,27]. GVHD scoring according to the National Institutes of Health criteria was not possible because of the retrospective nature of the study.

The secondary outcomes were overall survival (OS), defined as days to death from any cause or last follow-up, and relapse-related mortality (RRM), defined as days to death due to relapse or last follow-up. Viral infection–reactivation was defined as any virus antigen positivity reported in blood, urine, stool, or swabbed body surface. A further outcome of interest was immune reconstitution, defined as the repopulation of CD4+ T-lymphocytes. A CD4+ T-lymphocyte count of at least 500 cells/μL in two consecutive measurements within 100 days after HSCT was considered successful immune reconstitution. Patients who died before 100 days of follow-up were assessed until the date of death.

Thymic reconstitution was evaluated by MRI assessment of the thymic volume at one, three, and twelve months from HSCT and RTE quantification at 100 days and one year from HSCT. The absolute RTE count was assessed with flow cytometry and defined as naïve T-lymphocytes expressing CD45+CD3+CD4+CD45RA+CD31+. To compare the post-transplant thymic volume with the thymic volume in the immunologically competent cohort of children without a record of hemato-oncological, inflammatory, or infectious diseases aged from 0 to 18 years (control group), we selected 250 thoracic MRIs performed for orthopedic or neurological reasons at our Institute. We calculated the thymic volume for all chosen records, and the measurements obtained were stratified for 18 age groups (each year corresponds to a single group). For an additional perspective, we also selected the patients aged ≤ 18 years who, in the same period, underwent their first allogeneic HSCT after myeloablative conditioning without ATG (ATG-unexposed group) and compared the differences between their thymic reconstitution and that of the control group.

Statistical Analysis: Descriptive statistics were used to summarize patient baseline characteristics and determine the distribution and frequency of the variables. Fisher exact and chi-square tests were used to compare categorical variables, and the Wilcoxon rank-sum test was used to compare continuous variables.

Continuous variables were expressed as the median and confidence interval (CI) between the second and third quartiles (percentile 25 and percentile 75), while categorical variables were expressed as a frequency, as an absolute value, or as a percentage. Box and whisker plots were generated to display the distribution of the numeric variable. The Mann–Whitney test was used to compare different groups of patients as appropriate. The two-tailed Fisher exact test assessed the association between categorical variables. The OS and RRM were estimated using the Kaplan–Meier method. Kaplan–Maier curves were compared using the log-rank test. *p*-values < 0.05 were considered statistically significant. Statistical analyses were performed using WinStat (v.2012.1; In der Breite 30, 79189 Bad Krozingen, Germany) and MedCalc (Statistical Software version 18.9.1, Ostend, Belgium; http://www.medcalc.org; 2018, accessed on 13 July 2021).

## 3. Results

A total of 176 patients underwent HSCT at our center from April 2005 to April 2020. After exclusions, there were 102 patients in the ATG-exposed group and 69 patients in the ATG-unexposed group. There were no significant differences in sex, mean age at transplant, indication for HSCT, and disease risk index in the two groups. The distributions of patients in both groups were similar concerning conditioning regimen and graft source. Baseline patient characteristics in both groups are summarized in Table 1.

To investigate how ATG exposure can affect transplant-related outcomes, we further split the patients of the ATG-exposed group into a low ATG/lymphocyte ratio subgroup (ratio < 0.01) and a high ATG/lymphocyte ratio subgroup (ratio > 0.01). This decision was based on recently published data demonstrating the importance of the total lymphocyte count at the start of ATG infusion and the ATG cumulative dose [28,29]. The patients of both subgroups were comparable in terms of sex, age at transplantation, and disease risk index (Table 2).

The total lymphocyte count before ATG infusion was significantly higher, and the total ATG dose infused was significantly lower in the subgroup with a ratio < 0.01 in comparison to the subgroup with a ratio > 0.01 (*p* < 0.001; Table 2).

### 3.1. GVHD Incidence

A total of 36 patients in the ATG-exposed group (29 [59%] in the low ratio subgroup vs. 7 [16.6%] in the high ratio subgroup) developed some grade of acute GVHD (*p* < 0.001). The overall the 100-day cumulative incidence of grade I–II acute GVHD was 34.4% in the low ratio subgroup vs. 9.5% in the high ratio subgroup (*p* < 0.001), while grade III–IV acute GVHD occurred in 24.5% and 7.1%, respectively (*p* < 0.05; Figure 1). No difference between the two subgroups was observed in the cumulative incidence of chronic, limited, or extensive GVHD (*p* = 0.096).

### 3.2. Survival and Relapse

The low ratio subgroup had significantly higher OS compared to OS in the high ratio subgroup (80.4% vs. 58.0%, *p* = 0.031), while in contrast, non-relapse mortality (NRM) was significantly higher in the high ratio subgroup (13.1% vs. 33.0%, *p* = 0.035). There were no differences in event-free survival (64% in the low ratio subgroup vs. 47% in the high ratio subgroup) and RRM (5.9% vs. 9.8%) between the two groups (*p* = 0.11 and *p* = 0.7, respectively). These data, including the different causes of death in the two subgroups, are shown in Table 2. OS curves for patients in the two subgroups are presented in Figure 2. Multivariate Cox regression analysis confirmed the results obtained from the univariate analysis, namely that the primary diagnosis, type of conditioning, and hematopoietic stem cell source were not risk factors for OS and that only the ratio of ATG/lymphocytes (*p* < 0.03, HR = 2.4, 95% CI 1–5.4) was a risk factor (Table 3).

### 3.3. Hematopoietic Recovery and Virus Reactivation

No difference in hematopoietic recovery times was found between the two subgroups. The median time for neutrophil engraftment was 16 days (±5.6) in the low ratio subgroup vs. 15 days (±6.0) in the high ratio subgroup (*p* = 0.23) and 23 days (±7.9) vs. 25 days (±6.1), respectively, for platelet engraftment (*p* = 0.19).

Table 2 shows the two subgroups’ cumulative incidence of viral infections (human cytomegalovirus, Epstein–Barr virus, and adenovirus reactivation and disease). We did not find a significant difference in the cumulative incidence of virus reactivation between the low ratio and the high ratio subgroups (*p* = 0.06) or in the number of reactivation episodes. The difference was significant only for cytomegalovirus infection (39.2% in the low ratio subgroup vs. 60.7% in the high ratio subgroup, *p* = 0.043).

### 3.4. T-Cell and Thymic Volume Reconstitution Following HSCT

To correctly interpret the data, we compared the duration of post-transplant immunosuppression in both subgroups. The number of patients who were off immunosuppression at one year following HSCT was significantly larger in the high ratio subgroup (63% in the low ratio subgroup vs. 91% in the high ratio subgroup; *p* < 0.001).

The ATG/lymphocyte ratio during conditioning was predictive of the successful immune reconstitution of CD4+ T-cells. Table 2 summarizes the thymus-independent and thymus-dependent T-cell reconstitution data in the low and high ratio subgroups. The CD4+ count 100 days after HSCT was significantly higher in the low ratio subgroup (*p* = 0.0018), with this also being reflected in the speed with which the count of CD4+ T-cells reached > 500 cell/μL (130 days vs. 199 days, *p* = 0.043).

Similarly, the RTE count within one year after HSCT was considerably higher in the low ratio subgroup than in the high ratio subgroup (*p* < 0.001). We compared the RTE count within one year after HSCT of the two ATG-exposed subgroups and the ATG-unexposed group (Figure 3). As we predicted, the thymus-dependent T-cell reconstitution, defined as the number of RTEs at one year since HSCT, was significantly better in the ATG-unexposed group (*p* < 0.0001).

We calculated the thymic volume one year after HSCT for all patients in the ATG-exposed and ATG-unexposed groups, comparing them with the thymic volume of the control group made up of an immunologically competent age-matched pediatric cohort (Figure 4). The median thymic volume in the low ratio ATG-exposed subgroup was significantly higher than in the high ratio subgroup (14.7 cm^3^ vs. 4.5 cm^3^; *p* < 0.001; Table 2). No statistically significant differences in thymic volume were found between the low ratio ATG-exposed subgroup and the ATG-unexposed group (*p* = 0.082; Figure 4A). Moreover, the thymic volume of the control group was higher compared with ATG-exposed and ATG-unexposed transplant recipients (*p* < 0.0001; Figure 4B).

We analyzed a correlation between the thymic volume and the RTE count in the two ATG-exposed subgroups and the ATG-unexposed group one year after HSCT (Figure 5). A strong correlation (*p* < 0.0001) was found between the thymic volume and the number of RTEs in the two ATG-exposed subgroups (Figure 5A,B), while no such correlation (*p* = 0.1411) was found in the ATG-unexposed group (Figure 5C).

## 4. Discussion

Our study in a pediatric population retrospectively investigated HSCT-related outcomes concerning the intensity of ATG pre-transplant exposure and its relationship to post-transplant thymic volume restoration and thymus-dependent T-cell reconstitution. Finding the correct dose of ATG, especially in the pediatric population, should be a primary goal in optimizing its biological effect in terms of both averting GVHD and minimizing its impact on post-transplant immune reconstitution and, consequently, minimizing the infection rate [30,31].

Considering the limitations of a retrospective study, our data show that exposure to ATG does not derive from the cumulative dose alone but also from the total lymphocyte count at the onset of ATG infusion. We have shown that a higher dose of ATG associated with a lower lymphocyte count on the first day of ATG administration negatively influences the OS. Indeed, our data show that post-transplant T-cell subset reconstitution and thymic volume restoration were worse in the high ATG/lymphocyte ratio subgroup compared with the low ratio subgroup. Previous studies have shown how higher serum levels of ATG associated with low lymphocyte counts in the first week after HSCT are correlated with poor immune reconstitution and a higher susceptibility to infections [28,32]. Unfortunately, we have no data on the ATG serum concentration of our patients. Until very recently, no applicable pharmacokinetics and pharmacodynamics analyses were carried out at our center.

Previously published data show that low total lymphocyte counts are able to bind only part of the infused ATG, resulting in an extended persistence of free ATG in the recipient’s blood [29]. Our data confirm the widespread evidence that standard empiric weight-based ATG dosing that does not consider the pre-treatment lymphocyte count could lead to a very different ATG blood concentration compared with the concentration obtained when taking both the patient’s weight and pre-treatment lymphocyte count into account [33].

In this study, we compared the incidence of acute and chronic GVHD between low and high rate groups. Greater exposure to ATG appears to impact the incidence of acute GVHD but not chronic GVHD, in contrast to the data published by other research groups [10,34,35,36,37]. These latter studies were conducted in adult patients, which probably explains the higher incidence of chronic GVHD than in our results.

The low ATG/lymphocyte ratio group had better outcomes with respect to OS and NRM associated with a lower incidence of bacterial, fungal, opportunistic, and CMV-related infections. These findings confirm the randomized controlled trial and observational studies data on the infectious-related safety of moderate ATG exposure [8,38,39,40].

Our analysis did not find statistical significance in the incidence of relapse or RRM between patients in low and high ATG/lymphocyte subgroups, confirming data from previously published studies [41].

The innovative aspect of our study analyzed the influence of ATG exposure on the restoration of the thymic volume and, consequently, thymopoiesis recovery. Myeloablative conditioning is more frequently used in pediatric transplants. One of the most damaging consequences of myeloablative regimens is prolonged and profound immune suppression, mainly involving thymopoiesis, which can last for more than two years following HSCT [42]. As the thymus represents the primary site of T-cell development, the optimal recovery of the thymic function is of primary importance for survival after HSCT [21]. Several factors that injure the thymus may affect the recovery of thymopoiesis after HSCT: aging, acute and chronic GVHD, the source of stem cells, cytomegalovirus reactivation, or previous treatment of the underlying hematological malignancy [21,22,43]. Thymopoiesis is much more efficient in younger patients. The young thymus is able to fully recover its function when damaged by acute GVHD, reaching pretransplant values one year after the graft [44,45]. We compared the thymic volumes of all transplanted patients, ATG exposed and not, with the control group. Patients exposed to ATG and those not exposed one year after transplantation have a thymic volume that is statistically smaller than in healthy controls. These data are attributable to the conditioning-related damage of thymic tissue, GVHD, and the intensity of infectious complications due to immunosuppression. When comparing thymic volumes only between transplant patient groups, it appears that increased exposure to ATG plays an unfavorable role in thymic size recovery. We found no differences in thymic volumes between the unexposed ATG group and the low ATG/lymphocyte group. In contrast, in the high ATG/lymphocyte ratio group, the thymus size was significantly smaller in all age groups. Previously published studies have demonstrated that rabbit-derived ATG preparations exert a dose-dependent cytotoxic effect on primary human thymic stromal cells, particularly thymic epithelial cells [46]. Comparing RTE production at one year from HSCT in ATG-exposed and non-exposed patients, we found a significant delay in the ATG-exposed group. The high ATG/lymphocyte ratio group patients suffered a more significant delay in the reconstruction of thymopoiesis. Furthermore, we found a close correlation between the thymic volume and the circulating RTE values in this group, confirming the data reported by other studies.

Although the principal limitations of our study include the retrospective cohort and the heterogeneity of the primary disease, our results are strengthened by the comparability of the conditioning (myeloablative for all patients), GVHD prophylaxis, and supportive care.

Further studies are needed on individualized ATG dosing that does not detract from post-transplant immune renewal and that improves transplant-related outcomes in the pediatric population.

## Figures and Tables

**Figure 1 jcm-12-00730-f001:**
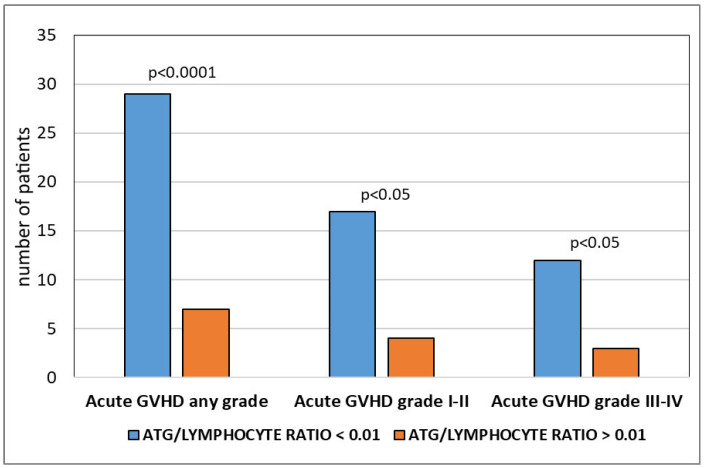
Cumulative incidence and incidence sorted by grades of acute GVHD in the low ATG/lymphocyte ratio group (blue columns) and high ATG/lymphocyte ratio group (orange columns).

**Figure 2 jcm-12-00730-f002:**
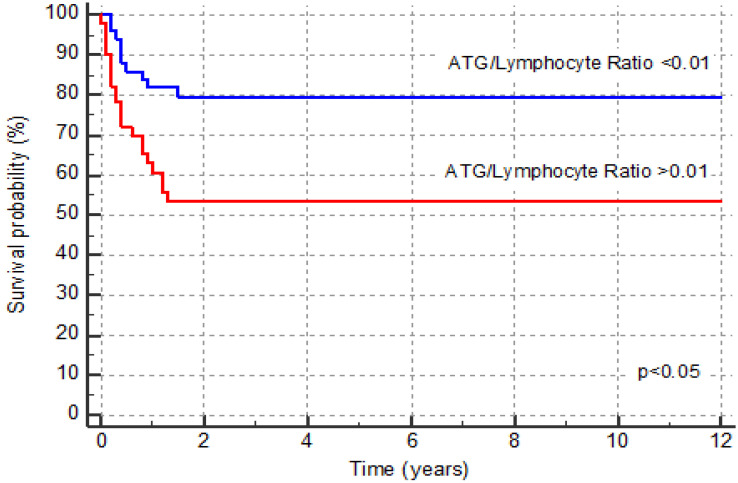
Kaplan–Meier curves for overall survival of patients in the low ATG/lymphocyte ratio group (blue line) and high ATG/lymphocyte ratio group (red line).

**Figure 3 jcm-12-00730-f003:**
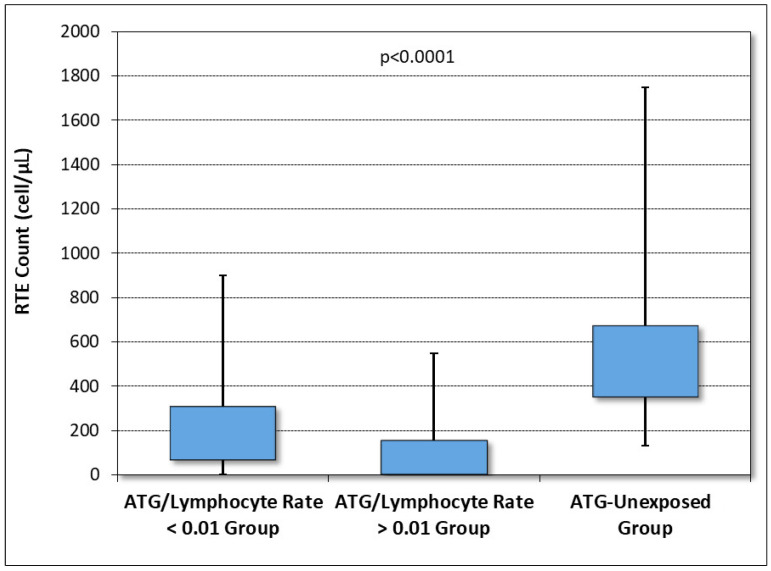
Box and whisker plot of recent thymic emigrant count (cell/μL) at one year after HSCT in the low ATG/lymphocyte ratio group (first box), high ATG/lymphocyte ratio group (second box), and ATG-unexposed group (third box).

**Figure 4 jcm-12-00730-f004:**
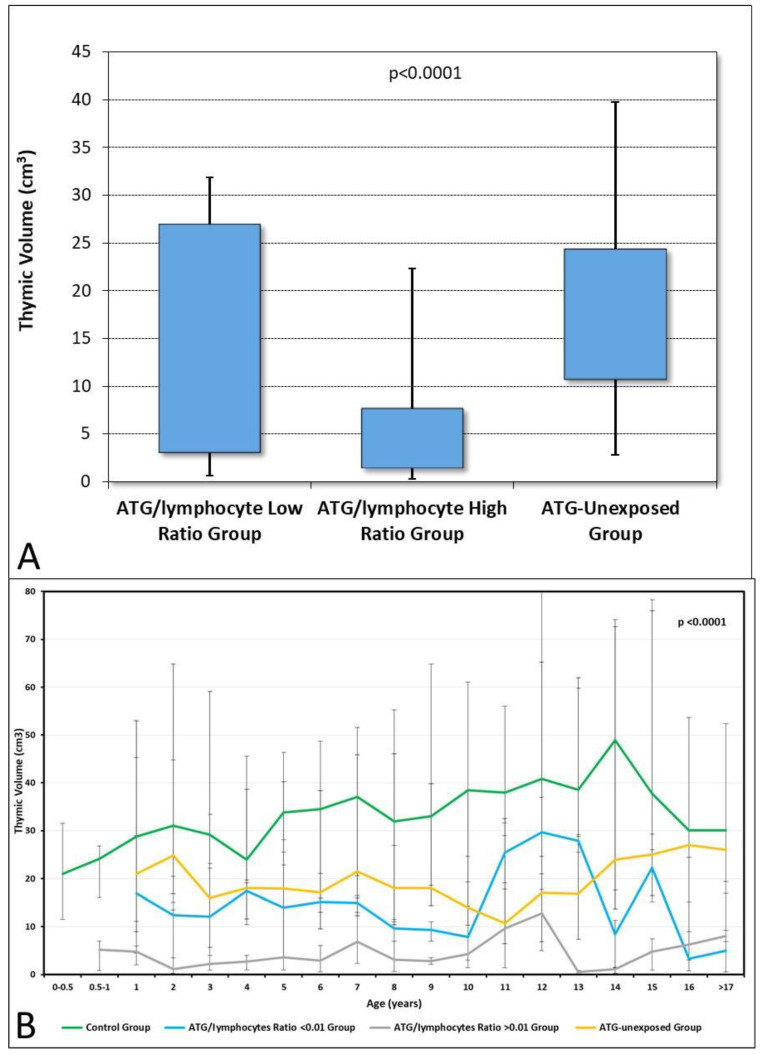
Thymic volume (cm^3^) at one year after HSCT. (**A**) Box and whisker plot of thymic volume in the low ATG/lymphocyte ratio group (first box), high ATG/lymphocyte ratio group (second box), and ATG-unexposed group (third box). (**B**) Line graph presentation of the age-matched thymic volume in the low ATG/lymphocyte ratio group (blue line), high ATG/lymphocyte ratio group (grey line), ATG-unexposed group (yellow line), and ontrol group (green line).

**Figure 5 jcm-12-00730-f005:**
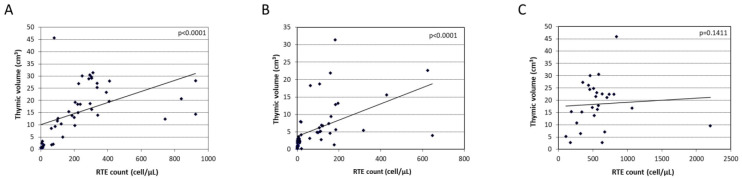
Python scatter plot of correlation between thymic volume (cm^3^) and recent thymic emigrant count (cell/μL) at 1-year post-transplantation. (**A**) Thymic volume and recent thymic emigrant count correlation in the low ATG/lymphocyte ratio group. (**B**) Thymic volume and recent thymic emigrant count correlation in the high ATG/lymphocyte ratio group. (**C**) Thymic volume and recent thymic emigrant count correlation in the ATG-unexposed group.

**Table 1 jcm-12-00730-t001:** Patient demographics and transplant characteristics of patients in low ATG/lymphocyte ratio group and high ATG/lymphocyte ratio group.

Pre-Transplant Characteristics	Study Group (ATG Exposed)	Control Group (ATG Unexposed)	*p*-Value
Number of patients (%)	102 (100)	69 (100)	
Sex:			
Male (%)	60 (58.9)	42 (60.9)	0.97
Female (%)	42 (41.1)	27 (39.1)	0.98
Age at transplant, years, mean (± SD)	8.25 (5.32)	9.03 (5.8)	0.86
Underlying disease, number (%):			
Acute lymphoblastic leukemia	39 (38.2)	25 (36.2)	0.87
Acute myeloid leukemia	19 (18.6)	14 (20.3)	0.81
Myelodysplastic syndrome	16 (15.6)	13 (18.8)	0.63
Solid tumor	5 (4.9)	2 (2.9)	0.62
Nonmalignant disorders	23 (22.5)	15 (21.7)	1
Disease risk index, number (%):			
Low	25 (24.5)	18 (26.1)	0.84
Intermediate	33 (32.4)	22 (31.9)	1
High	28 (27.4)	19 (27.5)	1
Very high	16 (15.7)	10 (14.5)	0.79
Myeloablative conditioning, number (%):			
MCHT-based	70 (68.6)	44 (63.8)	0.8
TBI-based	32 (31.4)	25 (36.2)	0.66
Graft source, number (%):			
Bone marrow	78 (76.5)	52 (75.4)	0.95
Peripheral blood stem cells	18 (17.6)	13 (18.8)	0.81
Umbilical cord blood	6 (5.9)	4 (5.8)	1
Allogeneic donor type, number (%):			
Matched related donor	6 (5.9)	69 (100)	<0.001
Matched unrelated donor	71 (69.6)	-	
Haploidentical donor	25 (24.5)	-	
Follow-up, weeks, median (range)	113 (2–679)	120 (4–703)	0.24

MCHT myeloablative chemotherapy. TBI total body irradiation. SD standard deviation. ATG anti-thymocyte globulin.

**Table 2 jcm-12-00730-t002:** Comparison of transplant outcomes between low ATG/lymphocyte ratio group and high ATG/lymphocyte ratio group.

Variables	ATG/Lymphocyte Ratio < 0.01 (*n* = 51)	ATG/Lymphocyte Ratio > 0.01 (*n* = 51)	*p*-Value
**Disease risk index, number (%):**			
Low/Intermediate	27 (53.0)	31 (60.8)	0.59
High/Very high	24 (47.0)	20 (39.2)	0.54
**Total ATG dose, mg/kg, patients (%):**			
<5	11 (42.3)	15 (57.7)	0.578
5–10	17 (48.6)	18 (51.4)	0.9
>10	23 (56.1)	18 (43.9)	0.58
**Lymphocyte count before ATG, cells × 10^2^/μL, mean (±SD)**	16.04 (13.2)	1.49 (1.3)	<0.001
**Total ATG dose, mg/kg, mean (±SD)**	6 (3.8)	13.2 (7.3)	<0.001
**CD4 day +100, cells/μL, mean (±SD)**	330 (187)	127 (97)	0.0018
**CD4 > 500 cells/μL, days, median (range)**	130 (0 *–1095)	199 (0*–450)	0.0431
**Recent thymic migrants + 1 year, cells/μL, median (range)**	206 (0 *–924)	59 (0*–647)	<0.001
**Thymic volume + 1 year, cm^3^, median (range)**	14.7 (0.4–45.7)	4.5 (0.2–31.4)	<0.001
**Acute GVHD (any grade), number (%):**	29 (59.0)	7 (16.6)	<0.001
Grade I-II	17 (34.4)	4 (9.5)	<0.001
Grade III-IV	12 (24.5)	3 (7.1)	<0.05
**Chronic GVHD, number (%)**	9 (18.4)	2 (4.8)	0.0965
**Virus infection/reactivation, number (%):**	28 (56.0)	38 (74.0)	0.06
CMV	20 (39.2)	31 (60.7)	0.0437
EBV	9 (17.6)	14 (27.5)	0.34
Adenovirus	5 (9.8)	5 (9.8)	1
<3 episodes	14 (27.0)	14 (27.0)	1
3–5 episodes	10 (19.6)	15 (29.4)	0.35
>5 episodes	5 (8.8)	9 (17.6)	0.38
**Endothelial damage-related complications, number (%) ****	3 (5.9)	9 (17.6)	0.12
**Off immunosuppression at 1 year, number (%)**	26 (63.0)	32 (91.0)	<0.001
**Overall survival, number (%)**	41 (80.4)	30 (58.0)	0.0313
**Event-free survival, number (%)**	33 (64.0)	24 (47.0)	0.11
**Non-relapse mortality, number (%):**	7 (13.1)	17 (33.0)	0.0357
GVHD	0	1 (2.0)	-
VOD	1 (2.0)	3 (5.9)	0.6
Infection	3 (5.9)	8 (15.6)	0.2
Other	3 (5.9)	5 (9.8)	0.7
**Relapse-related mortality, number (%)**	3 (5.9)	5 (9.8)	0.7

ATG anti-thymocyte globulin. SD standard deviation. GVHD graft versus host disease. VOD veno-occlusive disease. *CMV* cytomegalovirus. *EBV* Epstein–Barr virus. * Patients dead before the lymphocyte reconstitution. ****** Endothelial damage-related complications: VOD, capillary leak syndrome, engraftment syndrome, transplant-associated thrombotic microangiopathies, diffuse alveolar hemorrhage.

**Table 3 jcm-12-00730-t003:** Multivariate proportional hazards regression analysis for overall survival.

	Overall Survival	
Variable	Hazard Ratio (95% Cl)	*p*
**ATG/Lym Low** **< 0.01**	1	
**ATG/Lym High** **> 0.01**	2.4 (1–5.4)	0.03
**LLA**	1	
**LMA/MDS**	0.32 (0.08–1)	0.06
**Solid Tumor**	1.19 (0.2–5.4)	0.81
**Non Oncological Disease**	0.46 (0.13–1.6)	0.22
**TBI-based**	1	
**Chemo-based**	1.18 (0.67–3)	0.87
**BM**	1	
**PBSC**	0.81 (0.27–2.3)	0.55
**CB**	0.46 (0.6–3.6)	0.46
**MUD**	1	
**Haploidentical**	1.2	0.3

HR Hazard ratio. CI confidence interval. ATG anti-thymocyte globulin. AML/MDS acute myeloid leukemia/myelodysplastic syndrome. TBI total body irradiation. MCHT myeloablative chemotherapy. PBSC Peripheral blood stem cells.

## Data Availability

The data presented in this study are available on request from the corresponding author.

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
