# Peer review of "Pre-Transplant Total Lymphocyte Count Determines Anti-Thymocyte Globulin Exposure, Modifying Graft-versus-Host Disease Incidence and Post-Transplant Thymic Restoration: A Single-Center Retrospective Study"

_jcm, 2023, doi:10.3390/jcm12020730_

Round 1

Reviewer 1 Report

The study analyzes an important component of hematopoietic stem cell transplantation- the impact of serotherapy on the survival and the incidence of complications. The authors introduce the ATG/lymphocyte ratio and use it to separate patients into two groups, that show different outcomes.

My first remark is, that ATG/lymphocyte ratio was not properly defined- which values and units are used in nominator and denominator of this parameter?

Second, did the authors observe any correlation between ATG/lymphocyte ratio and total ATG dose? If yes, than the parameter can be supported as a substitute marker for ATG exposure and used in real life for adjustment of serotherapy dose.

In addition, I have got a question, what was the reason for differences in ATG dosage?

The data on the RTE reconstitution and thymic volume are interesting, but did their values show correlation with with patients' age? Using MRI for thymic volume assessment hasn't got practical value except for clinical studies.

The mansucript is otherwise very well prepared and the conclusions are supported by the results.

Author Response

On behalf of the authors, I thank the reviewers for their thoughtful and constructive comments. The authors have revised the manuscript to include further analyses and additional explanations. Enclosed is a detailed list of responses and changes made. All the changes in the manuscript have been highlighted in yellow. I hope that our endeavor to address these issues is satisfactory to both the editor and the reviewers. 

Sincerely,

Natalia Maximova

Reviewer 1

The study analyzes an important component of hematopoietic stem cell transplantation- the impact of serotherapy on the survival and the incidence of complications. The authors introduce the ATG/lymphocyte ratio and use it to separate patients into two groups, that show different outcomes.

My first remark is, that ATG/lymphocyte ratio was not properly defined- which values and units are used in nominator and denominator of this parameter?

Thank you for your advice. We added the sentence with a better definition of the ATG/Lymphocyte ratio in the Materials and Methods chapter (lines 93 – 96).

Second, did the authors observe any correlation between ATG/lymphocyte ratio and total ATG dose? If yes, than the parameter can be supported as a substitute marker for ATG exposure and used in real life for adjustment of serotherapy dose.

Thank the reviewer for the interesting observation. In fact, we have not looked into the correlation between ATG/lymphocyte ratio and the total ATG dose because the ATG dose was standard for all patients, and ATG/lymphocyte ratio was calculated retrospectively.

It was recently found that clearance of ATG was mainly influenced by body weight (<40 kg) and absolute lymphocyte count (ref. 28). Therefore, dosing ATG at a fixed mg/kg dose for all weights and not taking the absolute lymphocyte count into account will still result in high variability in ATG exposures between patients, subsequently causing unpredictable T-cell recovery.

In addition, I have got a question, what was the reason for differences in ATG dosage?

The reason for differences in ATG dosage reported in our cohort was due to changes in institutional practice over the fifteen years covered by the study. For example, ATG dosage at the beginning of the study period was the same for all patients who received grafts from MUD or haploidentical donor independently of the primary disease. However, in the following years, patients with nonmalignant diseases, such as hemoglobinopathies, received a higher dosage than patients with leukemia who underwent intensive treatment. Then, in recent years, sickle cell patients receiving transplants from siblings have also received conditioning that included low- to medium-dose ATG.

The sentence on lines 89-91 summarizes the reason for the differences in ATG dosage.

The data on the RTE reconstitution and thymic volume are interesting, but did their values show a correlation with with patients' age? Using MRI for thymic volume assessment hasn't got practical value except for clinical studies.

Thanks for the question. We are particularly interested in this topic, so much so that we have a prospective ongoing study whose objective is the study of post-transplant morphological reconstruction of the thymus evaluated with MRI, and especially its correlation with the age of the patients.

First, the thymus volume is completely zeroed after myeloablative conditioning at any age, both infant and adult. Does thymus rebuilding after HSCT follow the donor's features? An adolescent thymus, after receiving neonatal stem cells, would increase its volume and activity? Similarly, how does an adult stem cell donor transplantation affect a neonate recipient's thymus?

Studying thymus volumes after immunological reconstruction following transplantation, we noted the following issues:

  • Regardless of the donor's age, the recipient's thymus reaches a complete reconstruction harmonically with chronological age. This appears to be particularly true in an infant transplanted for SCID.
  • A possible “hypertrophic” reconstruction in comparison to the recipient's age in an adolescent patient transplanted with cordon blood stem cells (or thymic hyperplasia rebound after high dose chemotherapy?).
  • The unchanged thymic volume before and after transplantation with an optimal immune system restoration as in couples adult donor and recipient.

We know that the study of thymic volume is impractical in routine, but it seemed important to make our data available to readers.

The mansucript is otherwise very well prepared and the conclusions are supported by the results.

Reviewer 2 Report

The manuscript by Grasso et al here descried a single-center retrospective study on the effect of ATG exposure on lymphocytes. The manuscript is nicely written. However, I have a minor comment that needs to be addressed before acceptance of the manuscript. 

In figure 4B, the author included a control group (green line) for comparison. More information needs to be added about this group. 

Author Response

On behalf of the authors, I thank the reviewers for their thoughtful and constructive comments. The authors have revised the manuscript to include further analyses and additional explanations. Enclosed is a detailed list of responses and changes made. All the changes in the manuscript have been highlighted in yellow. I hope that our endeavor to address these issues is satisfactory to both the editor and the reviewers. 

Sincerely,

Natalia Maximova

Reviewer 2

In figure 4B, the author included a control group (green line) for comparison. More information needs to be added about this group. 

We defined better the Control group in the Material and Method section (lines 129 – 133).